# Short Tandem Repeats (STRs) as Biomarkers for the Quantitative Follow-Up of Chimerism after Stem Cell Transplantation: Methodological Considerations and Clinical Application

**DOI:** 10.3390/genes11090993

**Published:** 2020-08-25

**Authors:** Almudena Navarro-Bailón, Diego Carbonell, Asunción Escudero, María Chicano, Paula Muñiz, Julia Suárez-González, Rebeca Bailén, Gillen Oarbeascoa, Mi Kwon, José Luis Díez-Martín, Carolina Martínez-Laperche, Ismael Buño

**Affiliations:** 1Department of Hematology, Hospital General Universitario Gregorio Marañón, 28007 Madrid, Spain; anavarrobailon@gmail.com (A.N.-B.); diegocarbonell3@gmail.com (D.C.); asun_suni@hotmail.com (A.E.); mariachicano06@gmail.com (M.C.); paulamuniz@gmail.com (P.M.); rebeca.bailen@gmail.com (R.B.); gillen.oarbeascoa@salud.madrid.org (G.O.); mi.kwon@salud.madrid.org (M.K.); jdiezm@salud.madrid.org (J.L.D.-M.); 2Instituto de Investigación Sanitaria Gregorio Marañón, IiSGM, 28007 Madrid, Spain; julia.suarez@iisgm.com (J.S.-G.); cmlaperchehgugm@gmail.com (C.M.-L.); 3Genomics Unit, Hospital General Universitario Gregorio Marañón, 28007 Madrid, Spain; 4Department of Medicine, School of Medicine, Universidad Complutense de Madrid, 28040 Madrid, Spain; 5Department of Cell Biology, School of Medicine, Universidad Complutense de Madrid, 28040 Madrid, Spain

**Keywords:** hematopoietic stem cell transplantation, chimerism, leukocyte lineages, STR-PCR, quantitative PCR

## Abstract

Chimerism refers to the relative proportion of donor and recipient DNA after hematopoietic stem cell transplantation (HSCT) and its quantitative follow-up is of great clinical utility in this setting. PCR of short tandem repeats (STR-PCR) constitutes the gold standard method for chimerism quantification, although more sensitive PCR techniques (such as qPCR) have recently arisen. We compared the sensitivity and the quantification capacity of both techniques in patient samples and artificial mixtures and demonstrated adequate performance of both methods, with higher sensitivity of qPCR and better quantification skills of STR-PCR. By qPCR, we then prospectively followed up 57 patients that were in complete chimerism (CC) by STR-PCR. Twenty-seven patients (59%) showed 0.1–1% recipient DNA in the bone marrow. Only 4 patients presented 0.1–1% recipient DNA in peripheral blood (PB), and one of them relapsed. Finally, by qPCR, we retrospectively studied the last sample that showed CC by STR-PCR prior to relapse in 8 relapsed patients. At a median of 59 days prior to relapse, six patients presented mixed chimerism by qPCR in PB. Since both approaches have complementary characteristics, we conclude that different techniques should be applied in different clinical settings and therefore propose a methodological algorithm for chimerism follow-up after HSCT.

## 1. Introduction

Allogeneic hematopoietic stem cell transplantation (HSCT) consists on the substitution of a damaged hematopoietic system of a patient (recipient) by a new one from a healthy donor. It constitutes the only curative treatment for many hematological diseases [1]. If HSCT is successful, hematopoietic cells will be donor-derived, while the rest of the cells will be of recipient-origin. The coexistence of cells from different genetic origins within the same organism is known as a biologic chimera [2]. Therefore, the assessment of the proportion of DNA that belongs to donor and recipient after HSCT is named chimerism analysis [3]. The situation in which only donor-DNA is detected in a post-transplant sample is named complete chimerism (CC), while the detection of both donor and recipient DNA is referred to as mixed chimerism (MC). Chimerism can be studied in peripheral blood (PB) or bone marrow (BM) samples, as well as in specific leukocyte subsets, purified using immunomagnetic technology. The latter allows to significantly increase the sensitivity of the detection of donor and/or recipient DNA and to obtain more precise information on the biologic mechanisms underlying the clinically relevant processes that take place after HSCT. In the first few weeks after transplant, chimerism assessment helps physicians to assess engraftment [4,5,6], i.e., the correct recovery of hematologic counts in PB, with dominance of donor over recipient cells. An early acquisition of CC often precedes the development of graft versus host disease [7,8,9], which is of great interest for the clinical management of patients. At last, the reappearance of recipient cells in a patient that had previously achieved CC can help diagnose or anticipate relapse of the underlying disease [6,10,11]. In the follow-up of a treated neoplastic patient, when the number of malignant cells is so low that it can only be detected by highly sensitive techniques, it is called minimal residual disease (MRD). Although there are techniques for the detection of MRD that provide both higher sensitivity and specificity, they are not available for every patient, and chimerism can be used as a surrogate analysis. In the context described, chimerism analysis is of crucial importance for the follow-up of transplanted patients, since it can drive the implementation of immunomodulatory measures for the management of different post-HSCT complications, which are more effective when performed in an early manner [11,12,13].

Several biomarkers have been proposed for chimerism quantification. To date, the gold standard technique is short tandem repeat PCR (STR-PCR) [14,15]. STR-PCR has well-proven quantification capacity, which makes it more suitable for early post-transplant monitoring, but it has a sensitivity of 1–5% [14,16,17]. Although this sensitivity threshold can be lowered by studying chimerism in leukocyte subsets, it still may be insufficient for patients in later phases after transplantation, for the early diagnosis of complications (i.e., disease relapse). Therefore, quantitative PCR (qPCR) techniques have been introduced lately [18,19,20]. Analysis by qPCR targeting insertion deletion polymorphisms (indel) is an easy and well-known technique, which achieves better sensitivity (0.01–0.1%) in PB and BM in comparison with STR-PCR [18,20,21,22,23,24]. Nevertheless, the clinically significant threshold of recipient cell percentage is not yet well established, as well as the quantification capacity in comparison with STR-PCR, especially when high recipient cell percentages are present [21,25,26].

The aim of this study was to compare the technical efficacy and clinical utility of two chimerism quantification techniques: the gold-standard conventional STR-PCR and a new approach based in qPCR targeting indel polymorphisms.

## 2. Patients and Methods

### 2.1. Patients and Samples

The present study was approved by the local Ethics Committee (Code PI17/1880) and was carried out in accordance with The Code of Ethics of the World Medical Association (Declaration of Helsinki). Therapeutic approaches, sampling and diagnostic procedures were performed after patients and donors gave written informed consent.

For the correlation study, we retrospectively selected 171 samples from 24 patients with several chimerism determinations showing different percentages of recipient cells. Besides, in order to test the performance of the technique in diverse cell settings, samples stemming from BM, PB and separated cell lineages were selected (82 BM, 67 PB, 17 T-cells, 2 myeloid cells, 2 CD34-cells, 1 NK-cells).

In addition, two sets of 9 artificial mixtures were created using PB cells from two healthy subjects (a male and a female) with known percentages of male (putative recipient) leukocytes: 75%, 50%, 25%, 10%, 5%, 3%, 1%, 0.1%, 0.01%. Analysis on artificial mixtures was performed at least twice with each technique.

Between January 2017 and December 2019, all patients with hematologic malignancies and BM involvement that were in CC by STR-PCR were prospectively switched to qPCR for chimerism follow-up. A total of 57 patients were included. Patient characteristics are summarized in Table 1.

Additionally, to study the utility for anticipating relapse, we retrospectively selected 8 patients with hematological malignancies and BM infiltration that were in CC by STR-PCR prior to relapse. Patient characteristics are summarized in Table 2.

### 2.2. Genotyping

For leukocyte lineage analysis, we purified T lymphocytes (CD3^+^), myeloid cells (CD15^+^/CD33^+^) and NK cells (CD56^+^) from the PB, as well as stem cells (CD34^+^) from the BM of patient samples containing 10^7^ white blood cells. Cells were purified by immunomagnetic means (AutoMACS, Miltenyi Biotec, Bergisch Gladbach, Germany) using antibodies against each lineage marker. The minimum purity of isolated leukocyte subsets was 95%, as determined by flow cytometry. Cell lysates obtained from purified cell lineages by overnight incubation at 56 °C with proteinase K (100 μg/mL) were directly used for PCR.

For unseparated BM and PB samples, total genomic DNA was purified using Maxwell 16 Blood DNA Purification kit (Promega, Madison, WI, USA) following the instructions of the manufacturer. 

Chimerism analysis was performed by STR-PCR (Mentype^®^Chimera^®^ Biotype, Dresden, Germany) as a gold-standard comparative method and by indel quantitative PCR (Mentype^®^ DIPscreen, Mentype® DIPquant, Biotype, Germany), using at least two markers for each patient.

The STR-PCR kit employed included 12 polymorphic, autosomal, non-coding STR loci (D2S1360, D3S1744, D4S2366, D5S2500, D6S474, D7S1517, D8S1132, D10S2325, D12S391, D18S51, D21S2055 and SE33, see Appendix A) as well as the gender-specific marker Amelogenin (see Supplementary Information and Appendix A for further details) labeled on three different colors. PCR was performed with 0.2–1 ng of genomic DNA using the Mentype^®^Chimera^®^ PCR amplification kit (Biotype, Germany) and GeneAmp^®^ PCR System 9700 Thermal Cycler (Applied Biosystems, Foster City, CA, USA), with subsequent capillary electrophoresis in a Genetic Analyzer 3130*xl* (Applied Biosystems) under conditions recommended by the manufacturer. 

The qPCR kit included screening for 33 indel polymorphisms and the gender-specific locus Amelogenin (Mentype^®^DIPscreen multiplex PCR, Biotype, Germany). PCR was performed in a GeneAmp^®^ PCR System 9700 Thermal Cycler (Applied Biosystems), and capillary electrophoresis in the Genetic Analyzer 3130*xl* (Applied Biosystems). Once recipient-specific alleles were identified, qPCR was performed in patient samples using the Mentype^®^DIPquant kit (Biotype, Dresden, GmbH) in the LightCycler^®^ 480 Instrument II thermocycler (Roche Molecular Systems, Mannheim, Germany). 

All analyses were performed using Chimeris^TM^ Monitor Software (Biotype, Dresden, Germany), following the manufacturer’s instructions.

### 2.3. Data Analysis

CC was defined as any detection below 1% for STR-PCR according to the sensitivity of the technique. For qPCR, the CC threshold was set at 0.1%, based on prior reports in which lower recipient DNA percentages do not correlate with disease relapse [25]. Any detection of recipient’s DNA above these thresholds was defined as MC.

Statistical analyses and graphs were generated using Prism 8 for Windows (Versjon 8.4.2, GraphPad Software, LLC, San Diego, CA, USA).

## 3. Results

### 3.1. Correlation Analysis

Chimerism quantification of the 171 patient samples and 18 artificial mixtures was similar when comparing both techniques. The Bland and Altman analysis showed a small bias (−0.84), but differences were greater, with several results (dots) outside the 95% limit of agreement, with higher values of recipient cells (Figure 1A).

Due to the higher sensitivity of qPCR, of the 189 samples analyzed, 52 were positive and 15 negative by both methods (Figure 1B), while 122 were positive only by qPCR. Most of the samples in which recipient DNA was detected showed quantification of less than 1%, which corresponds to the sensitivity threshold of STR-PCR.

### 3.2. Analysis of Artificial Mixtures

Analysis of artificial mixtures (Figure 2A) provided evidence of significantly (≥2 logs) higher analytical sensitivity of qPCR (0.01%) over STR-PCR (1%). Regarding quantification capacity, for putative recipient’s DNA below 25%, both techniques performed similarly well. Nevertheless, for higher percentage of putative recipient’s (male) DNA, quantification capacity of STR-PCR was better than that of qPCR. In this setting, standard deviation (SD) was 11 and 14 for qPCR versus 4.3 and 2.9 for STR-PCR for known percentage of recipient’s DNA of 50% and 75%, respectively (Figure 2B). 

### 3.3. Informative Loci

We compared the number of informative loci by STR-PCR and qPCR in 57 patients. There were more informative (able to discriminate between donor and recipient) alleles for STR-PCR than for qPCR, when normalized to the number of alleles screened (12 and 33, respectively). Therefore, the mean percentage of informative loci was 31% for STR-PCR and 20% for qPCR (Figure 3), which is concordant with previous publications [26]. 

### 3.4. Implementation of qPCR for Follow-Up of Patients in Complete Chimerism

Chimerism analysis by qPCR was prospectively performed in 57 patients over a two-year period. A summary of patient characteristics is shown in Table 1. All patients were initially followed up by STR-PCR to assess engraftment and were switched to qPCR at a median of 13 months. Of note, 22 patients were transplanted before 2017 and started qPCR follow-up from January 2017. We performed qPCR in PB of all patients and in BM of 81% patients. Low levels (0.1–1%) of recipient DNA were detected in PB in only 4 patients (7%), while in BM in 27 patients (59%). Despite the high proportion of patients with detectable recipient DNA in BM, only one patient relapsed, and relapse was anticipated by a positive result targeting an MRD molecular marker (WT1 overexpression). Interestingly, this patient also presented quantification of >0.1% recipient DNA in PB by qPCR prior to relapse. Taken altogether, more than half of the patients presented low levels of recipient DNA in BM samples, which does not correlate with disease relapse. However, patients with detection of recipient DNA in PB might be at higher risk of relapse.

### 3.5. Detection of Impending Relapse

We sought to evaluate the possibility of anticipating relapse by analyzing the last sample in CC measured by STR-PCR prior to relapse diagnosis. We quantified chimerism by qPCR in 8 patients with hematologic malignancies and medullary infiltration (Table 2). Samples were collected at a median of 59 days prior to relapse (range 7–363). Most of the samples were positive at low levels (0.1–1%). Only two samples were negative: Patient 1, which was collected almost a year before relapse, and Patient 8, who presented an isolated extramedullary (testicular) relapse. Patient 1 had a positive result for the molecular MRD marker (WT1) at the time of disease relapse, while in Patient 8, the molecular MRD marker remained negative in the last sample and also at relapse.

## 4. Discussion

Quantitative follow-up of the chimerism status is essential for the management of patients after HSCT. The perfect technique for chimerism analysis should be cheap, fast, easy to perform and interpret and applicable to every patient. In addition, the results should be highly reproducible, with low variability between samples. This is especially important in the early phases after transplant, in which it is crucial to know the exact quantification of donor/recipient DNA to assess donor engraftment and diagnose graft failure in an early manner. Besides, the technique should also be applicable in different leukocyte subsets, such as T-cells, as this provides further information on the clinical outcome. Finally, it should provide high sensitivity, so that early relapses can be detected mostly in later phases after the transplant. Within this scenario, we will now discuss the differences between STR-PCR and qPCR, both in technical aspects and in clinical interpretation, and propose a methodological follow-up algorithm, based on our experience.

### 4.1. Comparison between STR-PCR and qPCR: Technical Aspects

Because of the intrinsic characteristics of the techniques (conventional PCR versus qPCR) and of the polymorphisms used (STRs versus indels), each assay performs differently across several aspects. A comparison of the DNA amount needed, total turnaround time, labor intensity and material cost is presented in Table 3. 

Regarding DNA requirements, one of the main disadvantages of qPCR is the large amount of DNA needed (250 ng/well × 6 wells), in comparison with 0.2–1 ng for STR-PCR. Furthermore, for qPCR, 750 ng of pretransplant sample needs to be included as a positive control in each follow-up experiment. In this regard, it is not uncommon that patient follow-up is transferred to a different transplant center due to personal or clinical reasons. In these cases, it is not possible to perform qPCR follow-up, unless a pretransplant DNA sample is provided by the center of origin, as it is needed in each assay as positive control. In contrast, there are several solutions for STR-PCR follow-up when a pretransplant sample is not available: (1)The center of origin can provide baseline STR genotyping (when the same commercial kit is used).(2)Donor is usually available for DNA sample extraction and recipient is available for buccal swab DNA extraction. A post-transplant buccal swab is not suitable for recipient DNA sampling for qPCR purposes, since donor leukocytes may contaminate epithelial DNA and therefore, MC would be detected due to the high sensitivity of the technique. This is not an issue for STR-PCR, because it is easy to compare donor and recipient buccal swab samples and neglect those peaks that correspond to donor alleles, so that recipient peaks can be readily identified.(3)If only donor or recipient is available for DNA extraction, post-transplant samples could even be searched for recipient-derived peaks, based on the low probability (even in the case of transplantation from sibling donors, Figure 3) that donor and recipient share identical STR genotyping (3.3 x 10^−12^ for two randomly selected Caucasian individuals, as per user instructions of the manufacturer of a standard STR-PCR kit).

Similarly, the use of pretransplant DNA as a control for each experiment with qPCR raises the concern of pretransplant DNA sample exhaustion. In our experience, the average total DNA in a pretransplant sample yields around 15 μg, which would be exhausted after 20 follow-up tests. Tyler et al. [26] showed that although this issue is unlikely, the calculated reference (∆Cq) can be reused from the previous experiment with no significant differences if pretransplant DNA is exhausted. Bach et al. [23] demonstrate that qPCR can be performed with up to four-fold reduction of input DNA (15 versus 60 ng) without altering PCR efficiency. Nevertheless, the performance of the qPCR assay makes it necessary to have an easily reachable stock of pretransplant DNA samples for all patients that undergo HSCT. Likewise, the amount of DNA needed for qPCR is sometimes difficult to achieve for patients in the first weeks after HSCT, when total leukocyte count is below 0.5/mm^3^ and total DNA after extraction ranges 0–1000 ng. For STR-PCR, the pretransplant sample is only used once for screening. 

A significant advantage of qPCR is the turnaround time, with the possibility of having a report within a few hours from reception of sample. Analysis by STR-PCR takes longer until a final laboratory report is released, due to the necessity of performing capillary electrophoresis to the PCR products while in qPCR, quantification is performed at the same time PCR is running. For the same reason, a conventional PCR thermocycler and a genetic analyzer for capillary electrophoresis are required to perform STR-PCR, while for qPCR, only a specific real time thermocycler is needed.

Hands-on time is similar for both techniques when few samples are analyzed. Nevertheless, when samples from several patients are processed at a time (i.e., in a center with high flow of transplants), the pretransplant sample from each patient must be sought, and multiple master mixes need to be prepared, as a specific PCR will be run for each patient (see example of plate setup in Figure 4). This increases the possibility of technical mistakes to occur, which, in the case of qPCR, are very difficult to detect, as the result will be interpreted as negative (CC) if a wrong marker is used.

Regarding the costs, the price for each test (one post-HSCT follow-up sample) with STR-PCR is around 35€/sample. For qPCR, although reagents cost around 10€ per well, as the assay is set in triplicate, and at least three markers are used for each patient (reference, marker 1, marker 2), plus the pretransplant sample that has to be included in each experiment (a total of 15 wells, see example of plate setup in Figure 4), the total price for one follow up sample is around 150€. On the other hand, the cost of pretransplant screening is similar for both techniques (around 70€ per donor/recipient pair).

Another practical issue of concern is chimerism follow-up, both to monitor engraftment early after transplant and to anticipate relapse, in patients that receive HSCT from different donors (two transplants from different donors, cord blood transplant with third-party Human Leukocyte Antigen (HLA)-mismatched donor, or double cord transplant). Because a biallelic marker is not able to differentiate between more than two individuals, it is very difficult to find enough markers when DNA from three different origins is present in a post-transplant sample [26]. As STR are highly polymorphic, in our experience, STR-PCR shows a good performance for the detection of DNA from three different individuals [27]. 

Regarding applicability, the number of informative markers varies logically depending on the number of markers used for screening. Our group and others [26] have seen a higher percentage of informative alleles with STR-PCR than with indel qPCR (Figure 3). Therefore, the presence of an indel in the recipient which is absent in the donor, ranges from 80% to 100% [20,21,26]. For STR-PCR, using a platform especially designed for chimerism analysis, the probability of finding an informative allelic constellation is above 99% [14].

In our study, the quantification capacity with high percentage of recipient cells is more accurate with STR-PCR than with qPCR. This observation has been previously reported by others [22,28], and it is explained by the variability of the qPCR method (up to one quantitation cycle). This variability results in errors of 0.5–2 times the true value, which become very large when a high percentage of recipient DNA is present. Because of this drawback, qPCR results are less reproducible than those obtained with STR-PCR, which makes it necessary to set the qPCR experiments in triplicate, and makes analyses more cumbersome, expensive and difficult to interpret. As the clinical interpretation of chimerism results is often based on the dynamics of several results across follow-up, quantification accuracy is of great clinical importance. 

The knowledge of intrinsic features of each technique helps with interpreting results in the clinical context, and also helps with choosing the correct technique depending on the center’s resources or on the patient’s chimerism status. Therefore, a qualitative evaluation of the performance features of each technique is summarized in Figure 5. 

### 4.2. Comparison between STR-PCR and qPCR: Clinical Aspects

Without any doubt, the most important advantage of qPCR over STR-PCR is sensitivity. On one hand, an increase in sensitivity up to 0.01% opens the possibility of widening the time period of anticipation of relapse, allowing for early intervention. On the other hand, with a higher sensitivity method, the number of false positive results increases. Several groups have found that the positive predictive value of qPCR is low, but the negative predictive value is around 99–100% [29,30]. This makes it difficult to set a clinically relevant threshold for which anticipation of relapse is likely. 

Regarding the lower sensitivity of STR-PCR, a strategy that has been proposed to overcome this problem is the study of chimerism in leukocyte subsets, separated by immunomagnetic technology. By analyzing chimerism status in leukocyte subsets, the sensitivity of the technique is highly increased, because it allows for detecting a very small amount of recipient cells, which remain unseen when diluted in the rest of the cell subsets [31,32]. Furthermore, it is well known that chimerism status within different cell subpopulations varies depending on the subpopulation studied (sometimes referred to as dissociate chimerism), as well as the clinical characteristics of underlying disease and transplant procedure, and gives additional information on outcome [5,10,27,33,34,35,36,37]. For example, the analysis of chimerism in leukemic lineages has demonstrated increased sensitivity and is directly related to the risk of disease relapse [10,12,36]. Our data suggest that qPCR is a reliable technique in terms of chimerism quantification in leukocyte subsets. However, it would be logical to think that, as the sensitivity of chimerism analysis in leukocyte lineages depends on the purity of immunomagnetic separation (5%), this effort might not be worthy, taking into account that the sensitivity of qPCR is 0.01% in total PB. Altogether, we think that there is enough evidence to continue performing analysis in leukocyte subsets, as it remains informative, although the most suitable technique for this analysis would probably be STR-PCR.

Another aspect that has raised interest with the use of qPCR is the detection of low levels of recipient DNA in patients that do not subsequently develop relapse. Like other authors [24,30,38], we think that this finding may correlate with the presence of residual healthy recipient hematopoietic cells or cells from non-hematopoietic origin, which would not be replaced by donor-derived cells after HSCT. This is especially relevant in BM samples, for which STR-PCR is known to provide higher sensitivity [39,40,41], but have also shown higher sensitivity when using qPCR [42]. In our experience, 59% of patients present detectable recipient DNA in BM at low levels (0.1–1%) at some point after HSCT, but this does not correlate with disease relapse. The detection of recipient DNA in PB is much less frequent and seems to be clinically useful in the anticipation of disease relapse, based on our results. Nevertheless, the establishment of cutoff points is still a matter of discussion nowadays, and careful interpretation of chimerism dynamics, together with other clinical data (such as MRD follow-up), are needed when using qPCR approaches.

Additionally, the lack of clinical utility of the result of a single data point has driven efforts to focus on the study of chimerism dynamics including several time points. In this setting, various studies have shown higher risk of relapse among patients that present progressive increase of recipient DNA over time [20,25].

### 4.3. Methodological Algorithm for Quantitative Chimerism Follow-Up

To date, few guidelines on chimerism monitoring have been reported. The gold standard technique is STR-PCR [11,14,15,43] but, after the publication of some of the studies discussed above, some centers have started to use qPCR for routine chimerism analysis. Regarding chimerism follow-up, there is still some controversy about the schedule and source of samples, the leukocyte subset to analyze and the need to perform chimerism studies in patients that undergo HSCT using myeloablative conditioning regimens, since most of them achieve CC rapidly [31]. There have been some efforts from single centers to publish recommendations based on their experience, as well as immunomodulation strategies depending on the results of chimerism analysis [5,11,26,30]. Here, we present a series of recommendations on chimerism analysis based on our experience, summarized in Table 4.

STR-PCR is a simple, cheap and reliable technique, which is applicable to almost every recipient–donor pair, even in the related (sibling or haploidentical) setting. Monitoring of engraftment in early phases after HSCT requires very high quantification capacity, and percentages of host DNA are expected to be high. As we mentioned before, leukocyte counts remain low until 2–4 weeks after HSCT, which makes it difficult to obtain high DNA amount after extraction. Besides, monitoring of T-cell chimerism is especially useful in this setting to anticipate graft failure or rejection [5,44,45]. For these reasons, we consider that chimerism analysis should be performed using STR-PCR from day 15 in unseparated PB and T-cells every other week (or before if clinically suspected graft failure) until achievement of CC. Because immunomodulatory strategies need to be implemented rapidly in case of impending graft failure, in our center, we perform sample extraction and processing (including immunomagnetic separation) on Monday and Tuesday, PCR on Wednesday and capillary electrophoresis on Thursday morning, so that the results are reported to clinicians on Thursday afternoon. Chimerism results are discussed weekly in the Hematopoietic Transplant Committee meeting early on Friday morning (Table 5).

Once CC and stable engraftment are achieved, monitoring should be performed according to clinical criteria based on the underlying disease. In the case of “non leukemic” diseases (i.e., lymphoma or multiple myeloma), chimerism analysis is unlikely to anticipate relapse, and main follow-up should be performed according to disease-specific follow-up (PET-CT, immunoglobulin electrophoresis, free light chains, etc.). We propose to perform chimerism analysis in PB using STR-PCR on months 1, 3, 6, 12 and 24 after HSCT. Chimerism monitoring can be eased thereafter.

For patients receiving HSCT for non-malignant diseases, high sensitivity is not needed, as the presence of recipient cells is frequent and often has a “protective effect” on the development of graft versus host disease, and in the majority of cases, a relatively small proportion of healthy hematopoietic cells can restore production of lacking cells or protein to a level that is enough to maintain a normal function. In such cases, we propose to use STR-PCR for chimerism analysis, with the same schedule than that for lymphoma or myeloma patients.

In the case of patients with “medullary” diseases (i.e., Acute and Chronic Leukemia, Myelodysplastic Syndrome, Myeloproliferative Neoplasm), chimerism analysis can help the early diagnosis of relapse. Although sensitivity with qPCR approaches can be close to that of MRD molecular markers and flow cytometry MRD analyses, the detection of recipient DNA does not necessarily translate the presence of malignant cells, as discussed above. However, despite MRD analyses being more specific than chimerism analyses, they can fail to detect relapse when expression of surface or molecular markers is lost due to clonal heterogeneity, which is not uncommon. Moreover, in the rare situation of donor cell leukemia [46,47,48], the presence of leukemic cells can be misdiagnosed as disease relapse if chimerism analysis is not performed. Therefore, we consider that monthly chimerism follow-up should be performed in unseparated PB in acute leukemia patients during the first year after HSCT and every three months during the second year. Because of the increase in sensitivity for relapse detection, in acute leukemia patients, we also perform chimerism analysis in unseparated BM (as well as in BM-isolated CD34^+^ cells in CD34^+^ neoplasms) every three months for the first year after HSCT and every six months for the second year. After the second year post-HSCT, we perform chimerism follow-up yearly in PB. Regarding the technique, for acute leukemia patients, in which an increase in sensitivity would have a direct benefit in relapse anticipation, we recommend planning follow-up depending on the availability of a reliable MRD marker. For patients with a sensitive MRD marker (i.e., BCR-ABL, NPM1, flow cytometry), we would encourage to perform STR-PCR complemented with MRD follow-up. In case of non-conclusive results or MRD positiveness in the limit of detection, qPCR could be performed to rule out relapse, as its negative predictive value is very high. We consider that qPCR follow-up can be useful in patients with lack of a MRD marker, or availability of a less sensitive marker. In these cases, results need to be interpreted taking chimerism dynamics and the clinical context into account. When an increase in host DNA is detected by qPCR, analysis should be repeated after one week to confirm the trend.

## 5. Conclusions

In conclusion, it is of crucial importance to know the intrinsic features of each technique for chimerism quantification, since both approaches can be complementary. Because of the higher quantification capacity with high recipient DNA percentage of STR-PCR, it could be more useful for early post-transplant monitoring, while highly sensitive qPCR could be used for relapse detection once CC is achieved. A deeper knowledge of these techniques can help us decide which one to use, depending on the individual clinical setting and taking into account the local resources and workflow organization.

## Figures and Tables

**Figure 1 genes-11-00993-f001:**
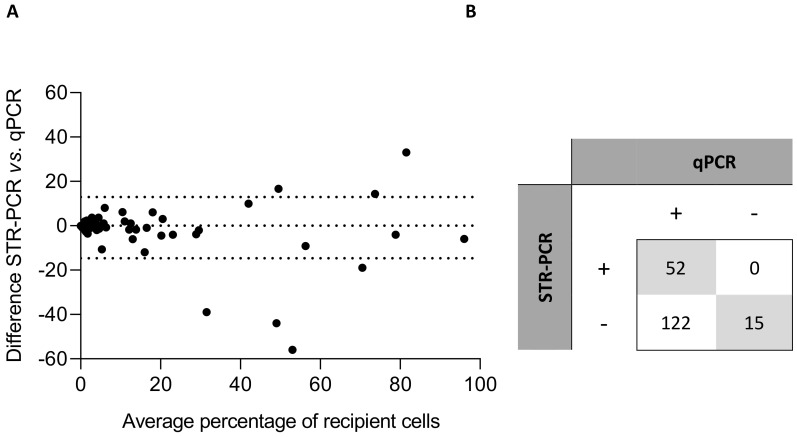
Correlation analysis. (**A**) Bland and Altman representation of the results of chimerism quantification with both techniques in 171 patient samples and 18 artificial mixtures. (**B**) Number of positive (presence of recipient DNA, MC) and negative (absence of recipient DNA, CC) samples with each technique. A segregated comparison of the results obtained in BM and PB samples is provided in Appendix A.

**Figure 2 genes-11-00993-f002:**
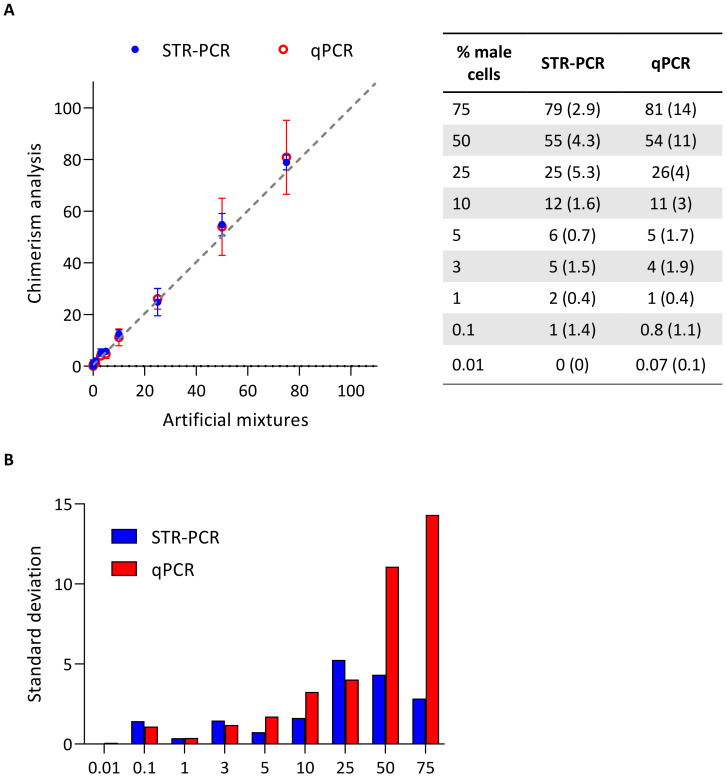
Artificial mixture analysis. (**A**) Results of chimerism quantification using STR-PCR (blue) and qPCR (red), versus actual percentage of cells of putative recipient (male). Results are represented as mean and standard deviation (SD, in brackets) of different independent experiments (n = 2 for STR-PCR, and n = 3 for qPCR). A representation using a logarithmic scale in the x and y axes is provided in Appendix A, in order to better discriminate differences in values below 10%, which are clinically less significant. (**B**) Standard deviation of both techniques by percentage of recipient cells.

**Figure 3 genes-11-00993-f003:**
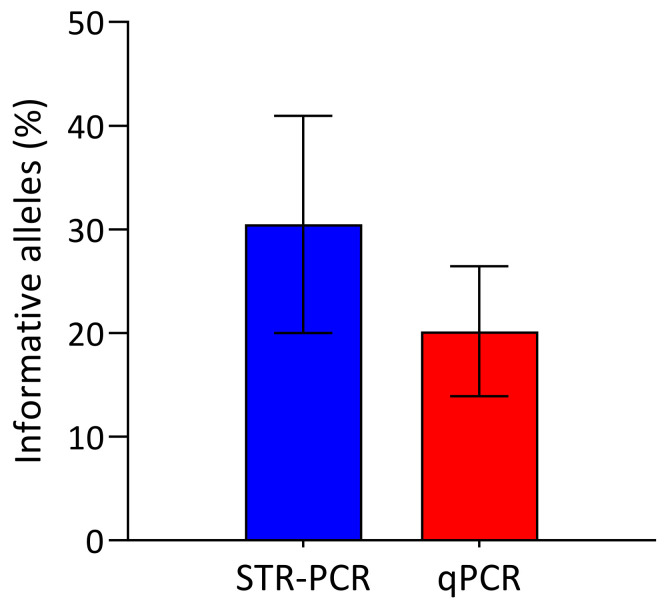
Informative loci. Percentage of informative loci for STR-PCR (blue) and qPCR (red).

**Figure 4 genes-11-00993-f004:**
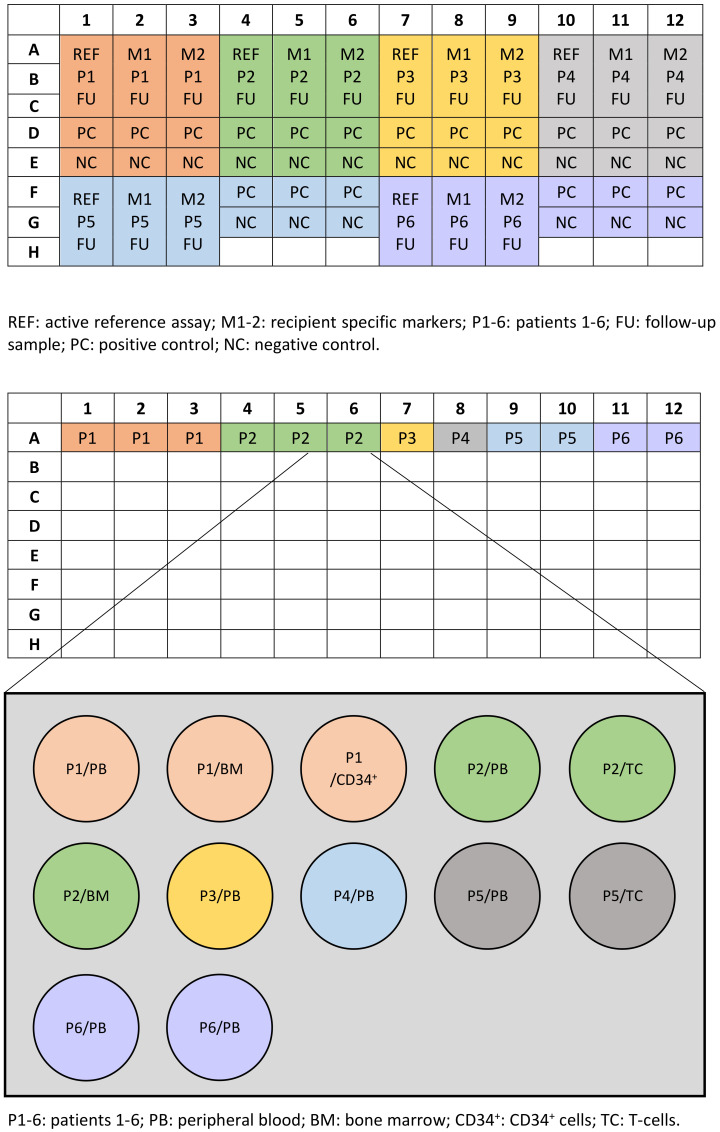
Example of plate setup for chimerism analysis of 6 patients with qPCR (top) and STR-PCR (bottom).

**Figure 5 genes-11-00993-f005:**
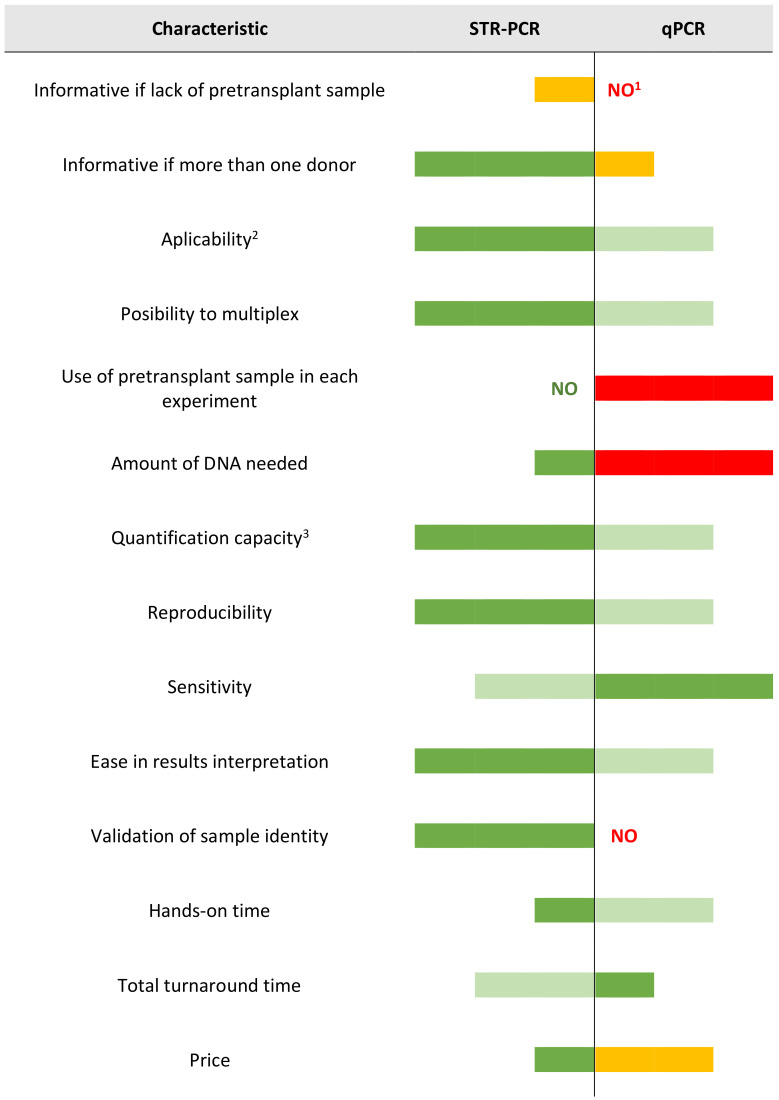
Comparison of technical characteristics of STR-PCR and qPCR. ^1^ Possible if monitoring for SRY gene in sex-mismatched donor/recipient pairs, by using any male donor as a positive control. ^2^ Percentage of informative loci per screened loci. ^3^ Especially with high percentage of expected minor allele. Colors in graph state for qualitative evaluation of each characteristic: very good (dark green), good (light green), bad (yellow), very bad (red). Length of bars state for quantitative evaluation of each characteristic (high, medium, low, none).

**Table 1 genes-11-00993-t001:** Patient characteristics, chimerism follow-up and outcome. ALL: Acute Lymphoblastic Leukemia; AML: Acute Myeloid Leukemia; MDS/MPN: Myelodysplastic Syndrome/Myeloproliferative Neoplasm; NHL: Non-Hodgkin Lymphoma.

Total–n	57
Sex Female—n (%)	20 (35)
Age, years—median (range)	46 (6–66)
Diagnosis—n (%)	
- AML	42 (74)
- ALL	10 (18)
- MDS/MPN	3 (5)
- NHL	2 (3)
Time to switch, months—median (range)	13 (1–48)
qPCR performed in PB—n (%)	57 (100)
Number of PB samples—mean (range)	6.12 (1–16)
Recipient DNA 0.1–1% in PB—n (%)	4 (7)
qPCR performed in BM—n (%)	46 (81)
Number of BM samples—mean (range)	2.14 (0–9)
Recipient DNA 0.1–1% in BM—n (%) - One sample - More than one sample	27 (59) 11 (24) 16 (35)
Molecular MRD marker—n (%)	
- None	13 (23)
- WT1	23 (40)
- NPM1	12 (21)
- Other	10 (18)
Positive molecular MRD marker—n (%)	2 (5)
Relapse—n (%)	1 (2)
Follow-up—median (range)	32 (9–67)

**Table 2 genes-11-00993-t002:** Summary of results of chimerism quantification by qPCR in relapsed patients. AML: Acute Myeloid Leukemia; MDS: Myelodysplastic Syndrome; RAEB2: Refractory Anemia with Excess Blasts type 2; CLL: Chronic Lymphocytic Leukemia; MF: Myelofibrosis; ALL: Acute Lymphoblastic Leukemia; CC: complete chimerism; MC: mixed chimerism; MM: molecular MRD marker.

Patient ID	1	2	3	4	5	6	7	8
Diagnosis	AML	MDS (RAEB2)	CLL	MF	AML	AML	ALL	AML
Relapse (days after HSCT)	1815	90	1703	253	479	322	218	363
Date of prior sample (days before relapse)	363	21	72	7	85	112	45	31
Type of prior sample	PB	PB	PB	PB	PB	PB	PB	BM
Result qPCR in prior sample	CC	MC	MC	MC	MC	MC	MC	CC
Quantification qPCR in prior sample (% recipient)	0.017	0.2	1.5	1	0.15	0.159	0.1	0.003
MM	WT1	None	None	None	NPM1	WT1	None	WT1
Result MM at relapse	Pos	-	-	-	Pos	Pos	-	Neg
Result MM in prior sample	Neg	-	-	-	Neg	Neg	-	Neg

**Table 3 genes-11-00993-t003:** Assessment of consumables, cost per sample and process duration of STR-PCR and qPCR.

	STR-PCR	qPCR
DNA needed	0.2–1 ng	2250 ng
Total turnaround time (10 samples)	2.5 working days	1 working day
Hands-on time (10 samples)	1.5 h	2.5 h
Analysis time (10 samples)	0.5 h	0.5 h
Cost per sample (€)	35€	150€
Devices needed	Screening and follow-up: PCR conventional thermocycler Genetic Analyzer for capillary electrophoresis	Screening: PCR conventional thermocycler Genetic Analyzer for capillary electrophoresis Follow-up: qPCR thermocycler

**Table 4 genes-11-00993-t004:** Proposed algorithm for chimerism follow-up. ^1^ Lymphoma, Non-malignant diseases. ^2^ Hematological malignancies with medullary infiltration (i.e., Acute and Chronic Leukemia, Myelodisplastic Syndrome, Myeloproliferative Neoplasm). CC: complete chimerism. GVHD: Graft *versus* Host Disease. MRD: Minimal Residual Disease.

Objective	Patients	Leukocyte Lineage	Technique	Chronogram
Engraftment	All	PB T-cells (B-cells, NK-cells)	STR-PCR	From day 15, every other week until CC (More frequently if needed)
		BM	STR-PCR	Day +30
GVHD	All	T-cells (Activated leukocytes)	STR-PCR	Every other week until CC
Follow up after CC	“Non leukemic” diseases ^1^	PB and BM	STR-PCR	Days 90, 180, 365
Follow up after CC—detection of relapse	“Medullary” diseases ^2^ with MRD marker	PB	STR-PCR	Monthly during first year, every 3 months for second year
		BM	STR-PCR	Every 3 months for first year
	“Medullary” diseases^2^ without MRD marker	PB	qPCR	Monthly during first year, every 3 months for second year
		BM	qPCR	Every 3 months for first year
	“Medullary” diseases^2^ CD34^+^	CD34^+^ from BM	STR-PCR	Every 3 months for first year

**Table 5 genes-11-00993-t005:** Week workflow for STR-PCR analysis.

Monday	Tuesday	Wednesday	Thursday	Friday
Collection of samples Immunomagnetic separation DNA extraction	Collection of samples Immunomagnetic separation DNA extraction	PCR	Capillary electrophoresis Data Analysis Report	Transplant Committee meeting

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
