# Peer review of "Short Tandem Repeats (STRs) as Biomarkers for the Quantitative Follow-Up of Chimerism after Stem Cell Transplantation: Methodological Considerations and Clinical Application"

_genes, 2020, doi:10.3390/genes11090993_

Round 1
Reviewer 1 Report
Paper well written providing very useful information for the hematology field in regards to detection and tracking of mix chimerism and its use for relapse detection. Also like the comparison done between the two techniques investigated (qPCR vs STR-PCR). Paper well written, however, at times too lengthy.
minor edits required:
Figure legend 1: indicate number of patient samples presented in figure in legend.
Fig 2A graph: could a log10 scale be used for both axes? that would stretch the data between 0.01 and 1% and better show the actual results.
Section 3.6 and 4.1 are interesting, however, they are a bit long, author should review and see if all text is really needed.
Figure 5: is there a rational for selection of color for the bar graphs? its not defined in the legend.
End
Reviewer 2 Report
In this paper, the authors compared two quantitative methods, namely the short tandem repeats (STRs) and quantitative PCR (Q-PCR) for measuring the chimerism after stem cell transplantation. The deep exploring of methodological comparisons and clinical implications strengthens the merits of the manuscript. However, several points may need to be further addressed as described below.
- The identity of STRs used in this paper seems to be obscure. Is it satellite DNA? Does it include any variants of different tandem repeats? Need to clearly specify.
- The quantitative comparison matters most in the scenario of minimal residual disease (MRD). It’ll be better to include a comparative form showing the detecting efficiency between STRs and Q-PCR specifically for MRD.
- Peripheral blood test is more acceptable for patients, however, many conclusions were drew from bone marrow. Adding a table specifying the peripheral blood vs bone marrow will help to address the point.
- Is there any time-course data for the HSCT patients? For example, from the MRD phase all the way to the relapse phase, showing the quantitative difference between STRs and Q-PCR?
